# Seq2seq is All You Need for Coreference Resolution

**Wenzheng Zhang**[1]     **Sam Wiseman**[2]     **Karl Stratos**[1]
[1] Rutgers University     [2] Duke University
{wenzheng.zhang, karl.stratos}@rutgers.edu
swiseman@cs.duke.edu

## Abstract

Existing works on coreference resolution suggest that task-specific models are necessary to achieve state-of-the-art performance. In this work, we present compelling evidence that such models are not necessary. We finetune a pretrained seq2seq transformer to map an input document to a tagged sequence encoding the coreference annotation. Despite the extreme simplicity, our model outperforms or closely matches the best coreference systems in the literature on an array of datasets. We also propose an especially simple seq2seq approach that generates only tagged spans rather than the spans interleaved with the original text. Our analysis shows that the model size, the amount of supervision, and the choice of sequence representations are key factors in performance.

## 1 Introduction

The seminal work by Lee et al. (2017) popularized end-to-end models for coreference resolution based on searching over all possible spans and their clustering. However, even with substantial refinement and simplification in followup works (Lee et al., 2018; Xu and Choi, 2020; Wu et al., 2020; Kirstain et al., 2021), the models are highly task-specific, involving many specialized hyperparameters such as how many candidates to retain in the mention proposal phase.

There is a recent line of works that take an alternative approach, leveraging advances in pretrained sequence-to-sequence (seq2seq) models. Liu et al. (2022) propose ASP, an autoregressive pointer-based model with a multitasking head for bracket pairing and span labeling. Bohnet et al. (2023) propose a transition-based system with carefully designed states and actions, simulated by a seq2seq model that processes one state at a time. However, these seq2seq-based models are *still* task-specific, requiring a modification of the seq2seq architecture or a derivation of a transition-based system with state manipulation.

A natural question is: are such task-specific models necessary for coreference resolution, or can we approach it as a standard seq2seq problem? There have been previous efforts to reduce coreference resolution as a seq2seq problem (Urbizu et al., 2020; Paolini et al., 2021), but they underperform task-specific models, suggesting that task-specific models are perhaps necessary.

In this work, we present the first full seq2seq reduction of coreference resolution that matches or outperforms the best coreference systems in the literature, demonstrating that task-specific models are not necessary to obtain state-of-the-art performance and questioning the need to develop task-specific solutions. Our approach is extremely simple. We treat the raw document as a source sequence and the coreference annotation as a target sequence, then finetune a pretrained encoder-decoder model like T5 (Raffel et al., 2020) or T0 (Sanh et al., 2022) without any modification to the architecture.

There is a great deal of flexibility in the choice of target sequence representation. Our main model represents the coreference annotation as a sequence of actions that either copy a token from the source sequence, start/end a mention span, or tag a predicted mention span with an integer. At test time, the model always produces a valid coreference clustering by constrained beam search. We consider an even simpler version in which the model generates only the tagged spans, and find that it also yields surprisingly high performance. This simpler version is advantageous because it results in faster inference.

Our seq2seq model obtains strong results on an array of coreference datasets. On English OntoNotes (Pradhan et al., 2012), with a 3B-parameter T0 the model obtains 82.9 test average F1, outperforming the corresponding ASP (Liu et al., 2022) initialized from the same base model (82.3). With a 11B-parameter T0, our model achieves 83.2 F1, outperforming CorefQA

(Wu et al., 2020) (83.1) and getting close to the best known result using a 13B-parameter model (83.3). On PreCo (Chen et al., 2018), the model achieves 88.5, outperforming the task-specific model of Toshniwal et al. (2021) (87.8). On Lit-Bank (Bamman et al., 2020), which has substantially smaller training data, the model obtains 78.3 cross-validation F1 lagging behind Toshniwal et al. (2021) who report 79.2. But when trained on the union of LitBank, OntoNotes, and PreCo, it obtains 81.2 split-0 F1, significantly outperforming their 78.2. Our analysis shows that the model size, the amount of supervision, and the choice of sequence representations are key factors in performance. We make our code publicly available at: `https://github.com/WenzhengZhang/Seq2seqCoref`.

## 2 Related Work

In this section, we give a more focused treatment of previous works on seq2seq-style approaches to coreference resolution to make our contributions more clear. Urbizu et al. (2020) propose framing coreference resolution as a seq2seq task, but their work is a proof of concept. They naively predict boundaries, cluster numbers, and blanks (e.g., "(0 − − 0) − (1 − (2) | 1)") with suboptimal modeling choices (e.g., their decoder only receives these symbols and no document content). They only report results on the ARRAU corpus and significantly fall behind existing works (e.g., 66.5 vs 78.8 under $B^3$). In contrast, we solve a much more challenging problem of developing state-of-the-art seq2seq coreference systems.

There are recent works approaching structured prediction with a general seq2seq-style solution. One example is TANL (Paolini et al., 2021), which frames entity/relation extraction, semantic role labeling, and coreference resolution as seq2seq tasks. Again, TANL fails to demonstrate competitive performance on standard coreference resolution datasets, obtaining only 72.8 average F1 on OntoNotes (compared to the current state-of-the-art performance level which is $> 80$). Their target sequence representation corresponds to our full linearization with token action except that they tag the cluster information by preceding mention strings (e.g., "[ his | Barack Obama ]"), which yields long target sequences and introduces clustering ambiguity. While we improve the performance of TANL to 79.6 in our own implementation, achieving the best performance (83.2) requires our sequence definitions (Section 3).

ASP (Liu et al., 2022) is an autoregressive pointer-based model for structured prediction. It is a modified transformer that at step $t$ conditions on the input document $x$ and a sequence representation of the annotation so far $z_{\leq t}$ and predicts a tuple of model-specific actions $(\alpha_t, \beta_t, \gamma_t)$ used for structure building. For instance, $\beta_t \in \{0 \ldots t-1\}$ is a bracket pairing action parameterized with a feedforward layer that consumes the previous hidden states $(h_t, h_{\beta_t})$ (i.e., a pointer network). ASP obtains state-of-the-art results on OntoNotes (up to 82.5 average F1). We outperform ASP with a standard transformer.

Bohnet et al. (2023) develop a transition-based coreference system that can be implemented by a seq2seq model. The system autoregressively maps a state to a prediction, where a state is previous coreference-annotated sentences along with the next sentence and a prediction is system actions (e.g., link and append). While the system can be reduced to seq2seq predictions, it processes one sentence at a time by applying the predicted actions to the current state. We show that such an explicit state-by-state transition is not necessary, and that a standard transformer can directly reason with all coreference clusters simultaneously.

## 3 Seq2Seq Methods

Let $\mathcal{V}$ denote the vocabulary. For any sequence $a \in \mathcal{V}^T$ and position $t \in \{1 \ldots T\}$, we use the prefix notation $a_{<t} = (a_1 \ldots a_{t-1}) \in \mathcal{V}^{t-1}$ and $a_{\leq t} = (a_1 \ldots a_t) \in \mathcal{V}^t$. We assume a generalized seq2seq setting in which $x \in \mathcal{V}^{T'}$ is the source sequence, $y \in \mathcal{V}^T$ is the target sequence (i.e., a sequence of labels), and $z \in \mathcal{V}^T$ is an additional sequence fed to the seq2seq decoder where $z_t = F(x, z_{<t}, y_{<t})$ is some fixed deterministic mapping. The "generalized" model uses parameters $\theta$ to define the conditional distribution

$$p_\theta(y|x) = \prod_{t=1}^{T} p_\theta(y_t|x, z_{\leq t}). \quad (1)$$

We emphasize that the generalization above does not modify the standard seq2seq framework. During training, we feed $(x, z)$ to the model as the usual source-target sequence pair and minimize the cross-entropy loss using $y$ as per-token labels. At test time, we apply the mapping $F$ at each step by postprocessing predictions.

All our methods use (1) and only change the variable definitions. The encoder input $x$ is always the document to be processed. The decoder input $z$ is a sequence representation or "linearization" of the coreference annotation of $x$. The decoder output $y$ is any action sequence from which $z$ can be extracted deterministically at each step.

## 3.1 Linearization of the Coreference Annotation

For a document $x \in \mathcal{V}^{T'}$, the coreference annotation $S$ is a set of spans clustered into $C$ groups formalized as

$$S \subset \big\{ (i, j, l) : \ 1 \leq i \leq j \leq T', \ 1 \leq l \leq C \big\}$$

Note that the spans can be nested. We assume that the spans are ordered in non-decreasing lengths. We will use the following as a running example:

$$x = (a, b, c, d, e)$$
$$S = \{(2, 2, 1), (5, 5, 2), (2, 3, 2)\} \qquad (2)$$

(i.e., the clustered spans are $\{\{b\}, \{(b, c), e\}\}$). The goal is to express $S$ as a sequence $z \in \mathcal{V}^T$ for some length $T$. A minimal formulation is to literally predict the integer triples, for instance $z = \mathtt{str}(S)$ where $\mathtt{str}$ is the string conversion in Python. However, while short, such a non-linguistic representation was found to perform poorly likely because it is not compatible with language model pretraining.

A better approach is to predict the mentions. We represent the tagged span $(i, j, l)$ in document $x$ as

$$\mathtt{<m>}\ x_i \ldots x_j \mid l\ \mathtt{</m>}$$

where $\mathtt{<m>}, \mathtt{</m>} \in \mathcal{V}$ are special symbols indicating the start and end of a mention representation, and $\mid\ \in \mathcal{V}$ indicates the end of a mention string. Nested spans are handled naturally by linearizing the spans in order (i.e., from the shortest to the longest). For instance, the subsequence $(b, c) \in \mathcal{V}^2$ in the running example (2) will become the length-10 sequence

$$\mathtt{<m>}\ \mathtt{<m>}\ b \mid 1\ \mathtt{</m>}\ c \mid 2\ \mathtt{</m>}$$

In contrast to a transition-based approach (Bohnet et al., 2023), we decode all coreference clusters in the document in a single pass. Thus there is an issue of alignment: if the model predicts multiple mentions with the same string form, how do we know

the corresponding spans in the source sequence? A simple solution popular in general seq2seq approaches to tagging is to exhaustively predict all the tokens in $x$ (Daza and Frank, 2018; De Cao et al., 2021). The example (2) is then linearized as

$$a\ \mathtt{<m>}\ \mathtt{<m>}\ b \mid 1\ \mathtt{</m>}\ c \mid 2\ \mathtt{</m>}\ d\ \mathtt{<m>}\ e \mid 2\ \mathtt{</m>}$$

We call this representation **full linearization**. Full linearization completely eliminates the problem of alignment ambiguity at the cost of longer target sequences. We also consider an alternative shorter representation that we call **partial linearization** in which only the tagged mentions are encoded. In this case, (2) is linearized as

$$\mathtt{<m>}\ \mathtt{<m>}\ b \mid 1\ \mathtt{</m>}\ c \mid 2\ \mathtt{</m>}\ \mathtt{<m>}\ e \mid 2\ \mathtt{</m>}$$

Partial linearization has the potential to drastically shorten the target length if the mentions are sparse, but it requires an explicit alignment step to transfer the predictions to the input document. We defer the discussion of the alignment problem to Section 3.4.

## 3.2 Action Sequences

Given a choice of linearization $z \in \mathcal{V}^T$ (i.e., input to the decoder), we can choose any action sequence $y \in \mathcal{V}^T$ (i.e., the actual predictions by the decoder) such that at each step $t$, we can extract $z_t$ from the document $x$ and the past information $z_{<t}$ and $y_{<t}$. We assume that $z_1 = \mathtt{}$ and $z_{T+1} = \mathtt{}$ are the standard start and end symbols for the target sequence. A straightforward action sequence is given by $y_t = z_{t+1}$ for $t = 1 \ldots T$ which we call **token action**. Token action corresponds to a common language modeling setting where the per-step prediction is simply the next token. For instance, the annotation $x = (a, b)$ and $S = \{(2, 2, 1)\}$ under full linearization and token action is assigned

$$y = (a, \mathtt{<m>}, b, \mid, 1, \mathtt{</m>}, \mathtt{})$$
$$z = (\mathtt{}, a, \mathtt{<m>}, b, \mid, 1, \mathtt{</m>}, \mathtt{}) \qquad (3)$$

Under full linearization, we can use a smaller action space $\mathcal{A} = \{\mathtt{<c>}, \mathtt{<m>}, \mathtt{</m>}, \mid, \mathtt{}\} \cup \mathcal{U}$ where $\mathtt{<c>}$ is the special "copy" symbol which means copying a single token from the document and advancing the index by one, and $\mathcal{U} \subset \mathcal{V}$ is the subset of the vocabulary used for encoding integers. We call this choice **copy action**, formally defined as

$$y_t = \begin{cases} \mathtt{<c>} & \text{if } z_{t+1} \notin \{\mathtt{<m>}, \mathtt{</m>}, \mid, \mathtt{}\} \cup \mathcal{U} \\ z_{t+1} & \text{otherwise} \end{cases}$$

The example (3) under copy action becomes

$$y = (\text{<c>}, \text{<m>}, \text{<c>}, |, 1, \text{</m>}, \text{})$$
$$z = (\text{}, a, \text{<m>}, b, |, 1, \text{</m>}, \text{})$$

Copy action in conjunction with constrained beam search is a natural way to prevent deviations between the source and target sequences (Daza and Frank, 2018). Partial linearization is not compatible with copy action since the model skips the gap between mentions.

### 3.3 Integer-Free Representation

So far we have relied on the separation symbol $|$ and an integer $l$ to label every mention with its cluster identity. Because the number of mentions in a document is quite large, we consider ways to avoid these two symbols. One way is to hard-code the cluster information directly in the mention boundaries by introducing a special symbol $\text{</m}_l\text{>} \in \mathcal{V}$ for each cluster number $l = 1, \ldots, C$. Under this scheme, we may assign to the annotation $x = (a, b)$ and $S = \{(1, 1, 1)(2, 2, 1)\}$:

$$y = (\text{<m>}, \text{<c>}, \text{</m}_1\text{>}, \text{<m>}, \text{<c>}, \text{</m}_1\text{>}, \text{})$$
$$z = (\text{}, \text{<m>}, a, \text{</m}_1\text{>}, \text{<m>}, b, \text{</m}_1\text{>}, \text{})$$

(4)

The performance of this representation was found to be surprisingly poor, likely because the model has a hard time learning to predict so many new symbols. We tackle this issue by introducing a "new" action $\text{<new>}$ that delegates the burden of specifying an unseen cluster number to postprocessing. The example (4) is now assigned the action sequence

$$y = (\text{<m>}, \text{<c>}, \text{<new>}, \text{<m>}, \text{<c>}, \text{</m}_1\text{>}, \text{})$$
$$z = (\text{}, \text{<m>}, a, \text{</m}_1\text{>}, \text{<m>}, b, \text{</m}_1\text{>}, \text{})$$

In this way, whenever the model predicts $\text{<new>}$ we can feed $\text{</m}_{l+1}\text{>}$ as the decoder input where $l$ is the previous number of clusters. The model is only responsible for predicting $\text{</m}_l\text{>}$ for expanding a known cluster $l$. We call this **integer-free representation** and show that it is nearly as performant as a corresponding baseline that relies on $|$ and $l$ while being shorter and notationally cleaner.

### 3.4 Mention Alignment

The issue of alignment arises only under partial linearization and token action. In this case, it is possible to have linearized mentions whose corresponding locations in the input document are ambiguous.

Consider the document $x = (a, b, c, d, e, b, b)$. The linearization consisting of two length-1 mentions

$$\text{<m>} \, b \mid 1 \, \text{</m>} \, \text{<m>} \, b \mid 1 \, \text{</m>} \qquad (5)$$

correspond to either $S = \{(2, 2, 1), (6, 6, 1)\}$ or $S = \{(6, 6, 1), (7, 7, 1)\}$. We align mentions by aligning tokens with gaps (i.e., a token may be aligned to nothing). Prior to aligning tokens, we remove all special symbols while saving the span information for each mention (omitting the cluster information for simplicity). For instance, (5) becomes $(b, b)$ with spans $(1, 1)$ and $(2, 2)$.

We then find a highest-scoring alignment where the score measures token matching (1 if matched, $-1$ if mismatched) and length-$n$ affine gap penalty $g(n) = -1 - p(n-1)$, where $p$ is a hyperparameter[1]. The affine gap penalty encourages long unsegmented gaps, capturing the intuition that mentions tend to appear in high density. The above example has the optimal alignment with 2 matches ($6 \leftrightarrow 1$ and $7 \leftrightarrow 2$) and a length-5 gap in the source sequence $(1 \ldots 5)$. Plugging in the token matches in the span information, we identify the locations $(6, 6)$ and $(7, 7)$ corresponding to the second annotation. This approach naturally handles nested mentions.

For a document of length $T'$ and a partial linearization (with special symbols removed) of length $K$, there are $\binom{T'+K}{K}$ possible alignments with gaps. We exactly find an optimal alignment in $O(T'K)$ time by Gotoh's algorithm (Gotoh, 1982). An oracle experiment shows that our approach is highly effective, reaching 98.0 average F1 on OntoNotes if we use the gold partial linearization. We further improve the alignment scheme by inserting sentence markers in the input document and linearization, which allows us to constrain the alignment to sentence pairs at test time.

## 4 Discussion

Having presented our technical approach, we discuss how it relates to other approaches to coreference resolution that also use sequence-based models.

Our approach is *pure* seq2seq because we use *both* the standard architecture and the system design that is applicable to any seq2seq task, such as machine translation and text summarization. In contrast, the approach of Bohnet et al. (2023) is not

---

[1]In our experiment, we set $p = 0$ for simplicity, as we observed negligible performance differences when $p \leq 0.0001$.

considered pure seq2seq because their transition-based system is specifically designed for the coreference task (e.g., their "Link" and "Append" actions are meant to cluster mentions together), even though the system itself is implemented using the standard seq2seq architecture. To understand the practical difference, note that their system requires $M$ encoder forward passes for a single document during inference where $M$ is the number of sentences, whereas ours requires one. The success of the transition system of Bohnet et al. (2023) does not imply the success of the general seq2seq system in our work.

We adopt the generalized seq2seq framework that may require postprocessing during generation (e.g., for copy action) for improved modeling flexibility and performance, but the postprocessing step does not change the standard seq2seq architecture and system design. Furthermore, our model *without* postprocessing is nearly as effective. Specifically, our "Full linear + token action + T0$_{3B}$" model in Table 2 is a standard seq2seq model with no postprocessing during inference but achieves an 82.4 test F1 score, which is competitive with our best same-size "Full linear + copy action + T0$_{3B}$" model that does require token-level postprocessing (82.9).

We use constrained decoding during generation to ensure valid coreference annotation, which is a standard practice (Daza and Frank, 2018; De Cao et al., 2021). The details can be found in Appendix A.

## 5 Experiments

### 5.1 Datasets

We train and evaluate on three widely used datasets for coreference resolution: OntoNotes (Pradhan et al., 2012), PreCo (Chen et al., 2018) and LitBank (Bamman et al., 2020). The data statistics are summarized in Table 1. It is important to note that these datasets exhibit significant variations in terms of size, document length, number of mentions/clusters, and domain. Following Kirstain et al. (2021), we incorporate the speaker's name into the text whenever there is a change in speakers for datasets that include speaker metadata. In addition to training and evaluating on these three datasets individually, we also perform an additional experiment involving joint training on the combined dataset comprising all three datasets. To address the issue of data magnitude imbalance in joint training, we adopt the methodology suggested by Toshniwal

| Dataset | # Docs | | | Words | Mentions | Cluster Size |
|---|---|---|---|---|---|---|
| | Train | Dev | Test | | | |
| OntoNotes | 2802 | 343 | 348 | 467 | 56 | 4.4 |
| LitBank | 80 | 10 | 10 | 2105 | 291 | 3.7 |
| PreCo | 36120 | 500 | 500 | 337 | 105 | 1.6 |

Table 1: Data statistics for OntoNotes, LitBank, and PreCo datasets. The number of documents in each split, average word count per document, average mention count per document, and average mention count per cluster are listed.

et al. (2021) and downsample the OntoNotes and PreCo datasets to 2K samples per epoch.

### 5.2 Implementation Details

We initialize our model using the T5 model family (Raffel et al., 2020), which includes models of various sizes. Specifically, we use T5 (Raffel et al., 2020), T0 (Sanh et al., 2022), and FLAN-T5 (Chung et al., 2022) with model sizes base, large, XL/3B, and XXL/pp. We use the pretrained models available in the Hugging Face Transformers library (Wolf et al., 2019).

To train large models with limited resources, we use Deepspeed (Rasley et al., 2020) with ZeRO optimizers (Rajbhandari et al., 2020) and enable gradient checkpointing. We divide the document into overlapped segments, each with a maximum length of 2048 tokens and an overlapping length of 1024 tokens. During inference, the maximum input length is 4096 tokens for all our experiments. We use constrained beam search with beam size 4.

For optimization, we use the Adam optimizer (Kingma and Ba, 2015) with the learning rate of 5e-4 for base/large models, 5e-5 for XL/3B, and 3e-5 for XXL/pp models. We use a linear learning rate decay scheduler with a warmup proportion of 0.1. We train our models using a batch size of 1 per GPU, using 8 A100 40G GPUs for models of size up to 3B and 8 A100 80G GPUs for models of size 11B.

### 5.3 Baselines

We categorize the baselines into the following three groups.

**Non-seq2seq** Models in this category are specifically designed for coreference resolution and employ sophisticated coreference-specific architectures. Most models in this category follow the approach introduced by Lee et al. (2017) to detect mention spans in the input text and establish

| | Model | MUC | | | B³ | | | CEAF$_{\phi_4}$ | | | Avg. |
|---|---|---|---|---|---|---|---|---|---|---|---|
| | | P | R | F1 | P | R | F1 | P | R | F1 | F1 |
| Non-Seq2seq | Lee et al., 2017 | 78.4 | 73.4 | 75.8 | 68.6 | 61.8 | 65.0 | 62.7 | 59.0 | 60.8 | 67.2 |
| | Lee et al. (2018) | 81.4 | 79.5 | 80.4 | 72.2 | 69.5 | 70.8 | 68.2 | 67.1 | 67.6 | 73.0 |
| | Joshi et al. (2019) | 84.7 | 82.4 | 83.5 | 76.5 | 74.0 | 75.3 | 74.1 | 69.8 | 71.9 | 76.9 |
| | Yu et al. (2020) | 82.7 | 83.3 | 83.0 | 73.8 | 75.6 | 74.7 | 72.2 | 71.0 | 71.6 | 76.4 |
| | Joshi et al. (2020) | 85.8 | 84.8 | 85.3 | 78.3 | 77.9 | 78.1 | 76.4 | 74.2 | 75.3 | 79.6 |
| | Xia et al. (2020) | 85.7 | 84.8 | 85.3 | 78.1 | 77.5 | 77.8 | 76.3 | 74.1 | 75.2 | 79.4 |
| | Toshniwal et al. (2020) | 85.5 | 85.1 | 85.3 | 78.7 | 77.3 | 78.0 | 74.2 | 76.5 | 75.3 | 79.6 |
| | Wu et al. (2020)* | 88.6 | 87.4 | 88.0 | 82.4 | 82.0 | 82.2 | 79.9 | 78.3 | 79.1 | 83.1 |
| | Xu and Choi (2020) | 85.9 | 85.5 | 85.7 | 79.0 | 78.9 | 79.0 | 76.7 | 75.2 | 75.9 | 80.2 |
| | Kirstain et al. (2021) | 86.5 | 85.1 | 85.8 | 80.3 | 77.9 | 79.1 | 76.8 | 75.4 | 76.1 | 80.3 |
| | Dobrovolskii (2021) | 84.9 | 87.9 | 86.3 | 77.4 | 82.6 | 79.9 | 76.1 | 77.1 | 76.6 | 81.0 |
| | Toshniwal et al. (2021) | - | - | - | - | - | - | - | - | - | 79.6 |
| | Liu et al. (2022) + T0$_{3B}$ | 85.8 | 88.3 | 86.9 | 79.6 | 83.3 | 81.5 | 78.3 | 78.5 | 78.4 | 82.3 |
| | Liu et al. (2022) + FLAN-T5$_{XXL}$ | 86.1 | 88.4 | 87.2 | 80.2 | 83.2 | 81.7 | 78.9 | 78.3 | 78.6 | 82.5 |
| Transition Seq2seq | Bohnet et al. (2023) + mT5$_{XXL}$ | 87.4 | 88.3 | 87.8 | 81.8 | 83.4 | 82.6 | 79.1 | 79.9 | 79.5 | 83.3 |
| Seq2seq | Paolini et al. (2021)+T5$_{base}$ | - | - | 81.0 | - | - | 69.0 | - | - | 68.4 | 72.8 |
| | Paolini et al. (2021)+T0$_{3B}^{\dagger}$ | 85.0 | 86.0 | 85.2 | 76.1 | 78.5 | 77.3 | 76.5 | 75.6 | 76.0 | 79.6 |
| | Partial linear + T0$_{3B}$ | 83.9 | 87.6 | 85.7 | 76.6 | 82.1 | 79.3 | 77.7 | 76.5 | 77.1 | 80.7 |
| | Integer free + T0$_{3B}$ | 84.9 | 88.8 | 86.8 | 78.9 | 84.0 | 81.4 | 78.1 | 79.3 | 78.7 | 82.3 |
| | Full inear + token action + T0$_{3B}$ | 85.9 | 88.6 | 87.2 | 79.6 | 83.5 | 81.5 | 78.9 | 78.0 | 78.5 | 82.4 |
| | Full linear + copy action + T0$_{3B}$ | 85.8 | 89.0 | 87.4 | 80.0 | 84.3 | 82.1 | 79.1 | 79.4 | 79.3 | 82.9 |
| | Full linear + copy action + T0$_{pp}$ | 86.1 | 89.2 | 87.6 | 80.6 | 84.3 | 82.4 | 78.9 | 80.1 | 79.5 | 83.2 |

Table 2: Results on the OntoNotes (CoNLL-12 English) test set. The average CoNLL F1 score of MUC, B³ and CEAF$_{\phi_4}$ is the main evaluation criterion. Models marked with † are our implementation. ∗ marks models using additional training data.

antecedent relationships by computing span representation similarity at either span level (Lee et al., 2017, 2018; Joshi et al., 2019, 2020; Xu and Choi, 2020; Liu et al., 2022) or word level (Kirstain et al., 2021; Dobrovolskii, 2021). In contrast, Wu et al. (2020) use a QA model to predict coreferent mention boundaries, while Xia et al. (2020), Toshniwal et al. (2020), and Toshniwal et al. (2021) use memory augmented models.

**Transition-based Seq2seq** Models in this category are based on a designed transition system. Currently Bohnet et al. (2023) is the only model in this category.

**Seq2seq** Models in this category leverage a sequence-to-sequence architecture to predict linearized coreference annotations without relying on coreference-specific model architectures or specialized system designs. Existing models in this category include Urbizu et al. (2020) and Paolini et al. (2021). However, Urbizu et al. (2020) do not evaluate on standard coreference resolution datasets like OntoNotes, and their performance is not competitive enough, so we do not compare with them. On the other hand, Paolini et al. (2021) do not report performance using larger T5 models, and their input length is shorter than ours (1024 words). To

ensure a fairer comparison, we implement their method and include the results by training and evaluating with a larger model (T0$_{3B}$) using the same input length as ours (2048 tokens). All of our models fall into this category. We include both full linearization models and partial linearization models as baselines. For full linearization models, we consider variants that use token action sequence and copy action sequence.

## 5.4 Results

### 5.4.1 English OntoNotes

Table 2 shows our main results on the test portion of English OntoNotes. We first point out that it is generally challenging to ensure full comparability due to numerous ways the approaches differ. For instance, CorefQA (Wu et al., 2020) achieves strong performance with a relatively small model (SpanBERT-large, 340M parameters), but it uses additional training data to optimize the mention proposal network (without which the performance drops to 75.9) and is slow at test time because it runs an extractive reader on each mention. On the other hand, the best results obtained with ASP (Liu et al., 2022) and the transition-based system (Bohnet et al., 2023) rely on much larger models (FLAN-T5$_{XXL}$, 11B parameters; mT5$_{XXL}$, 13B pa-

| Model | PreCo | LitBank | $LitBank_0$ |
|---|---|---|---|
| Xia and Van Durme (2021) | 88.0 | 76.7* | - |
| Thirukovalluru et al. (2021) | - | 78.4 | - |
| Wu and Gardner (2021) | 85.0 | - | - |
| Toshniwal et al. (2020) | - | 76.5 | - |
| Toshniwal et al. (2021) | 87.8 | 79.3 | 77.2 |
| Toshniwal et al. (2021)+Joint | 87.6 | - | 78.2 |
| Our copy action + $T0_{3B}$ | 88.5 | 78.3 | 77.3 |
| Our copy action + $T0_{3B}$ + Joint | 88.3 | - | 81.2 |

Table 3: Results on Preco, Litbank test set. The average CoNLL F1 score of MUC, $B^3$ and $CEAF_{\phi_4}$ is the evaluation metric. We report both the 10-fold cross-validation results (official setting) and the results of split 0 ($LitBank_0$) following Toshniwal et al. (2021). Joint denotes training on the union of OntoNotes, PreCo and $LitBank_0$. * marks transfer learning results which uses additional pretraining.

rameters), where running such large models is not always feasible depending on the authors' available computational resources (e.g., we do not have the resources to scale to 13B parameters). We do our best to interpret the results as objectively as possible[2].

The first observation is that models based on a sizeable T5 outperform those that are not (with the exception of CorefQA which uses additional training data). Focusing on models with 3B parameters and input length 2048, our T0-based seq2seq model with full linearization and copy action achieves an average F1 score of 82.9, outperforming ASP (82.3) and TANL (79.6) using the same base model and input length. In fact, our 3B-model outperforms ASP using a 11B model (82.5). While full linearization with copy action is the most performant, full linearization with token action (which requires no postprocessing during generation) is almost as competitive (82.4). Partial linearization with token action using the alignment method in Section 3.4 also obtains a nontrivial F1 of 80.7, outperforming most non-seq2seq baselines. In terms of performance and speed, partial linearization demonstrates lower accuracy but faster inference compared to full linearization. On the English OntoNotes development data, the inference time for partial linearization is about 22 minutes, whereas full linearization takes approximately 40 minutes for both token action and copy action sequences.

With the 11B-parameter $T0_{pp}$ initialization, our model reaches 83.2 F1, better than CorefQA and

close to 83.3 obtained by 13B-parameter transition-based system. While we do not have the computational budget to train a 13B-parameter model, as a point of comparison, Bohnet et al. (2023) report the dev average F1 of 78.0 using a 3.7B-parameter $mT5_{XL}$; our model using the same base model with full linearization and copy action obtains 79.6.

### 5.4.2 PreCo and LitBank

We further verify the effectiveness of our seq2seq model by taking the same 3B-parameter T0 setting (full linearization, copy action) for OntoNotes to additional datasets in Table 3. We consider PreCo (Chen et al., 2018) and LitBank (Bamman et al., 2020) following the same experimental setup in Toshniwal et al. (2021). On PreCo, which provides the largest training dataset (36K documents, compared to 2.8K in OntoNotes), our model outperforms all previous works at an average F1 score of 88.5.

LitBank, on the other hand, is a very small dataset with only 100 annotated documents split into 80-10-10 for training, validation, and test. Its official evaluation metric is an average over 10-fold cross-validation results; we also report results on split 0 to compare with prior work. Despite the small training data, our model achieves competitive results of 77.3 on split 0 and 78.3 on cross-validation, though lagging behind the task-specific model of Toshniwal et al. (2021). When the model is trained on the union of OntoNotes, Preco and LitBank split 0 training portion, the model achieves a significantly better performance of 81.2, beating 78.2 in the previous work. This shows that (1) when there is sufficient training data, our seq2seq model can easily obtain state-of-the-art results, and (2) even when that is not the case, the performance is still relatively strong and can be easily improved by including other available datasets.

### 5.5 Ablation Studies

In this section, we conduct ablation studies to investigate different aspects related to sequence representations, decoder input choices, and pretrained models of varying sizes. Unless specified, all the ablation experiments are conducted using the $T0_{3B}$ model.

### 5.5.1 Action Sequence

To analyze the impact of sequence representation, we present the results of an ablation study in Table 4.

---

[2]While the model sizes are similar, the comparison with Bohnet et al. (2023) is not fully equivalent due to their focus on multilingual coreference using the mT5 model.

| | Sequence Representation | Avg. F1 |
|---|---|---|
| full linear | Baseline | 82.6 |
| | – Copy action + integer free | 82.6 |
| | – Copy action + put integer before | 82.3 |
| | – Token action + integer | 82.5 |
| | – Token action + antecedent string | 79.5 |
| partial linear | Baseline | 81.1 |
| | – w/o sentence marker | 79.9 |
| | – Oracle align | 99.2 |
| | – Oracle align w/o sentence marker | 98.0 |

Table 4: Ablation study for sequence representations on OntoNotes development set. Average CoNLL F1 is reported.

**Full Linearization** The baseline for full linearization utilizes copy action sequence and represents cluster identity using integers. In Section 3.3, we introduce an integer-free representation which achieves performance comparable to the baseline (both achieving 82.6). Although the integer-free representation is more complex in design, it offers a cleaner and shorter sequence representation. Notably, placing the integer before the mention string leads to a noticeable drop in performance (82.3 vs 82.6), emphasizing the importance of predicting cluster identity after detecting mentions due to the autoregressive nature of the seq2seq model. Additionally, replacing the copy action in the baseline with a token action results in slightly worse performance (82.5 vs 82.6), indicating that a smaller action space is beneficial. Moreover, using the antecedent mention string to link coreferent mentions (similar to TANL (Paolini et al., 2021)) significantly decreases performance (79.5 vs 82.6). This demonstrates the superiority of using integers to represent cluster identity over using antecedent mention strings for linking coreferent mentions.

**Partial Linearization** Partial linearization is incompatible with the copy action since the model skips the gap between mentions. For partial linearization, the baseline employs a token action sequence with explicit sentence markers. Sentence markers prove to be useful for the model, allowing it to focus on each sentence individually during generation and aiding in alignment. Removing sentence markers leads to a significant deterioration in performance (79.9 vs 81.1). To further understand the benefits of sentence markers for alignment, we conduct oracle experiments using gold linearization with and without sentence markers, obtaining average F1 scores of 99.2 and 98.0, respectively.

| Decoder Input | Avg. F1 |
|---|---|
| Baseline | 82.6 |
| – Copy action sequence | 56.4 |
| – Token sequence + copy action sequence | 82.5 |

Table 5: Ablation study for decoder input on OntoNotes development set. Average CoNLL F1 is reported.

| Pretrained Model | # params | Avg. F1 |
|---|---|---|
| $T5_{base}$ | 220M | 76.2 |
| $T5_{large}$ | 770M | 77.2 |
| $T5_{3B}$ | 3B | 81.6 |
| $FLAN\text{-}T5_{XL}$ | 3B | 82.5 |
| $T0_{3B}$ | 3B | 82.6 |
| $FLAN\text{-}T5_{XXL}$ | 11B | 82.9 |
| $T0_{pp}$ | 11B | 83.0 |

Table 6: Ablation study for pretrained model on OntoNotes development set. Average CoNLL F1 is reported.

These results validate the effectiveness of sentence markers in alignment.

### 5.5.2 Decoder Input

We present an ablation study for decoder input in Table 5. The baseline uses a linearized token sequence as the decoder input. Replacing the token sequence with a copy action sequence (similar to Urbizu et al. (2020)) yields significantly worse performance compared to the baseline (56.4 vs 82.6). Averaging token and action embeddings as the input embedding is also less effective than the baseline (82.5 vs 82.6). These results emphasize the importance of providing the decoder with a linearized token sequence.

### 5.5.3 Pretrained Model

Table 6 shows an ablation study for pretrained model. We observe an improvement in performance as the model size increases. For models of the same size, both FLAN-T5 and T0 surpass the performance of the original T5 model. T0 achieves better performance than FLAN-T5 when compared at the same size.

### 5.6 Error Analysis

To better understand the model behavior, we conduct error analysis on the dev set in Table 7. The experiments are based on the $T0_{3B}$ copy action model. The unlabeled mention detection F1 is 89.2. On the other hand, the clustering performance reaches 95.8 average F1 when restricted to correctly recovered mentions, and 94.8 when we assume perfect mention detection. This shows that mention detec-

| | F1 |
|---|---|
| Mention detection | 89.2 |
| Detected mention clustering | 95.8 |
| Oracle mention clustering | 94.8 |

Table 7: Error analysis on OntoNotes development set. We report mention detection F1 and mention clustering average CoNLL F1.

tion is the performance bottleneck; once correct mentions are obtained, the model can accurately infer coreference clusters. Upon qualitative analysis of randomly sampled gold clusters and their best matches, we find that a major source of error is annotation mistakes. For instance, one gold annotation dictates "you do not want to [face]$_{17}$ the dilemma. But [it]$_{17}$ can not be avoided" while our model correctly predicts "you do not want to face [the dilemma]$_{20}$. But [it]$_{20}$ can not be avoided"; another gold annotation dictates "[ [ William ]$_9$ and she ]$_{10}$ saw each other, it was such a wonderful reunion for [ them ]$_{10}$ to just hug, and he would hug [ her ]$_2$ and look at [ her ]$_2$ " while our model predicts "[ [ William ]$_{11}$ and [ she ]$_2$ ]$_{12}$ saw each other, it was such a wonderful reunion for [ them ]$_{12}$ to just hug, and [ he ]$_{11}$ would hug [ her ]$_2$ and look at [ her ]$_2$".

## 6 Conclusions

We have presented a highly performant seq2seq reduction of coreference resolution. Unlike in previous works that rely on task-specific approaches to obtain strong performance, we use a standard encoder-decoder model that receives the document as an input sequence and predicts a sequence representation of its coreference annotation as the target sequence. Contrary to previously reported weak results using seq2seq reductions, we show for the first time that with suitable definitions of the sequence representation of the task, it is possible to achieve state-of-the-art performance, reaching 83.2 test average F1 on English OntoNotes with an 11B-parameter T0$_{pp}$ initialization. Our model's strong results fundamentally challenge the default task-specific approach to coreference resolution.

## Limitations

While our approach is standard seq2seq modeling and does not require architectural modifications, it requires an effort in identifying an effective sequence representation of coreference resolution; the model may perform poorly with a suboptimal choice of representation as evidenced in past works. However, once an effective representation is found, as is the case in this work, we expect the performance on the task will only improve with future advances in pretrained transformers. Another limitation, shared across all approaches built on seq2seq models, is the fact that training with language modeling loss is computationally intensive. Even with our substantial effort in scaling the model, with our budget we could not consider properly training models with more than 11B parameters. But we expect this limitation to benefit from the general effort in improving the efficiency of training and deploying transformers.

## Acknowledgements

We thank the anonymous reviewers for their helpful comments.

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

# A  Constrained Decoding

We implement constrained decoding through vocabulary masking, which depends on the current state of the generated sequence. Below, we outline the masking rules for various models. For more details, please check our code.

## A.1  Full Linearization

We categorize the state of the generated tokens thus far in full linearization with integer cluster identity as follows:

1. Inside Cluster Identity: Cluster identity generation stage (i.e., the count of mention end tokens </m> is less than that of separation token |).

2. Inside Mention: Open mentions exist (i.e., the count of separation tokens | is less than that of mention start tokens <m>), but not in Inside Cluster Identity state.

3. Outside: Not in Inside Mention or Inside Cluster Identity states.

**Token Action.** Depending on the current state, different tokens are permitted:

- Outside: The next token from the input source and the mention start token <m> are allowed.

- Inside Mention: The next token from the input source, the mention start token <m>, and the separation token | are allowed.

- Inside Cluster Identity: All integer tokens and the mention end token </m> are allowed.

**Copy Action.** For the Copy Action model, trained to generate a copy action token <c> rather than the actual next source token, we manipulate the logits scores to enforce the generation of the actual source token. Specifically, the logits score of the actual next source token is set to that of the copy token <c>. The copy token <c> is then masked out. The rules for permissible tokens remain consistent with those for Token Action.

## A.2  Partial Linearization

Due to alignment issues with the input source in partial linearization, it's infeasible to impose constraints on source input token generation as in full linearization. Nonetheless, sentence-level constraints can be applied by utilizing sentence boundary markers <sentence> and </sentence> in both the input source and linearization. We identify the current state of the generated tokens in partial linearization based on sentence markers and integer cluster identity as:

1. Complete Sentence: Equal counts of sentence start <sentence> and end markers </sentence>.

2. Inside Sentence $i$: Within the $i$-th sentence (i.e., <sentence> count is $i$ and not in Complete Sentence state).

3. Inside Cluster Identity: Cluster identity generation stage (i.e., the count of mention end tokens </m> is less than that of separation token |).

4. Inside Mention: Open mentions exist (i.e., the count of separation tokens | is less than that of mention start tokens <m>), but not in Inside Cluster Identity state.

5. Outside: Not in Inside Mention or Inside Cluster Identity states.

Depending on the current state, different tokens are permitted:

- Complete Sentence: only sentence start marker token <sentence> is allowed.

- Outside and Inside sentence $i$: All the tokens from the $i$-th sentence in the input source, the mention start token <m> and the sentence end marker token </sentence> are allowed.

- Inside Mention and Inside Sentence $i$: All the tokens from the $i$-th sentence in the input source, the mention start token <m>, the separation token | and the sentence end marker token </sentence> are allowed.

- Inside Cluster Identity and Inside Sentence $i$: All the tokens from the $i$-th sentence in the input source, all integer tokens, the mention end token </m> and the sentence end marker token </sentence> are allowed.

### A.3 Integer-Free Representation.

In the integer-free model, which is trained to predict <new> for unseen clusters instead of $</m_{l+1}>$, where $</m_0>, </m_1>, \ldots, </m_l>$ have already appeared, we manipulate the logits scores to enforce the generation of the $</m_{l+1}>$ token instead of <new> token for unseen cluster. Specifically, the logits score of the $</m_{l+1}>$ token is set to that of the <new> token. The <new> token is then masked out. We categorize the state of the generated tokens thus far in full linearization with integer-free cluster identity as follows:

1. Inside Mention Seen $l$: Open mentions exist (i.e., the count of cluster identity hard-coded mention end tokens $</m_l>$ is less than that of mention start tokens <m>) and $</m_0>, </m_1>, \ldots, </m_l>$ has been seen so far.

2. Outside Mention: No open mentions.

Depending on the current state, different tokens are permitted:

- Inside Mention Seen $l$: The next token from the input source, the mention start token <m>, and cluster identity hard-coded mention end tokens $</m_0>, </m_1>, \ldots, </m_l>, </m_{l+1}>$ are allowed.

- Outside Mention: The next token from the input source and the mention start token <m> are allowed.