# OpenReview forum: "Seq2seq is All You Need for Coreference Resolution"
_EMNLP/2023/Conference — EMNLP 2023 Main_

### Official Review · Reviewer_msDa · 2023-08-05

**Soundness:** 4

**Excitement:**

5: Transformative: This paper is likely to change its subfield or computational linguistics broadly. It should be considered for a best paper award. This paper changes the current understanding of some phenomenon, shows a widely held practice to be erroneous in someway, enables a promising direction of research for a (broad or narrow) topic, or creates an exciting new technique.

**Paper Topic And Main Contributions:**

This paper shows that linearizing coreference resolution problems and then using seq2seq models is highly performant. They argue that past models are all *task-specific* – i.e. having task-specific hyperparameters. Meanwhile, they present a method which linearizes the target sequence and guarantees well-formed output. They show competitive performance on several widely-used datasets and have ablations/analysis to explain the tradeoffs of certain annotation decisions and sources of errors.

**Questions For The Authors:**

First, this was a cool result! I had a couple questions.

* A. The claim throughout the paper is that model size has a big impact. This is neither surprising nor novel – for years, we’ve seen the base LM scale up for coref and it leads to (free) improvements on the downstream task. Table 6 claims that this is yet another example of this phenomenon. How do we know it is actually model size and not pre-training data? Both FLAN-T5 and T0 are trained on substantially more data. I don’t think there is any test set contamination, but do we actually think it is model size? If so, why are the 11B models only slightly better than the 3B ones?

* B. I offer a different hypothesis as to why seq2seq “suddenly” works now, and it isn’t entirely due to the linearization scheme (although that likely plays a big role). In Table 6, we see that T5_base/large do *worse* than SpanBERT-base/large or RoBERTa based models. This implies that had you done this study a few years ago, it would have been deemed unsuccessful. So why do you think seq2seq now works on the newer generation of models?
    * i. There were already some hypotheses put forth in the paper (like how it needs to be more natural-language), it would be nice to have them organized, as a discussion/retrospective section, as to why seq2seq should be the path forward.
* C. What do the sentence boundary markers look like? This isn’t mentioned in section 3.3/3.4 and it seems kind of important if it actually resolves many of the alignment errors.
  * i. If you had oracle alignment, (how) would the final score improve? It might be possible to “cheat” and coerce the intended alignment out of the model by looking at the attention weights in the final layer.
* D. The paper frequently emphasizes about not being “task-specific” – this doesn’t seem quite true and I suggest rephrasing this wording significantly? True to the definition in the paper, some training decisions still need to be made, like the hyperparameters for finetuning the T5/T0 model and max sequence length. These may need to be different depending on the size of the training set. Outside of this paper, a model that is not task-specific would be a general (coref) model that (at inference time) can predict properly on all tasks or datasets for the task, like a general "NLI" model or general QA model. If I understand the paper correctly, this paper is not claiming to be that omnibus model?
* E. The treatment for CorefQA in this paper might actually be a little unfair? How is their method more expensive than these 3B/11B models that required lots of extra pretraining for T5/T0? I know that for a long time, papers wrote them off as being unreproducible, unfair training conditions, “need TPUs”, etc. But given the sizes of the models in this work, is CorefQA finally comparable on paper?
* F. I'm still not clear on how <new> and <m_l> work. Does this imply that if a document has 100 clusters, there will be 100 special <m_l> tokens that are selectable by the decoder? I know this is rare, but there's no recency bias imposed to select between these 100, right?

**Reasons To Accept:**

* The paper is scientifically comprehensive and easy to follow; where possible, the authors tried to make fair comparisons.
* The authors give a timely demonstration that the seq2seq approach to coref, while not novel, can be highly competitive with much more room to explore.
* This simple approach also shifts the paradigm of “task-specific” coref models to more general models, where coref can simply be another dataset, fine-tunable with standard T5/T0 hyperparameters and methods.

Notably, even if there are major flaws/concerns with experimental design of the paper or reasons for the success of its method, it still presents a strong empirical result under the seq2seq paradigm, whereas many in the past have failed.

**Reasons To Reject:**

There are some minor terminology issues described below which make the comparisons appear fairer than they really are. Further, some of the conclusions drawn about why seq2seq now suddenly works need more justification, and a couple sentences don’t appear correct to me.

**Reproducibility:**

3: Could reproduce the results with some difficulty. The settings of parameters are underspecified or subjectively determined; the training/evaluation data are not widely available.

**Reviewer Confidence:**

5: Positive that my evaluation is correct. I read the paper very carefully and I am very familiar with related work.

**Typos Grammar Style And Presentation Improvements:**

* One of the limitations of this work is that it is only looking at English coreference. That’s fine, but Bohnet et al., 2023 is multilingual with mT5, and so that’s worth footnoting somewhere that even though model sizes are similar, comparison is still not entirely equivalent.
* It could be useful for reproducibility to include the constrained decoding rules/grammar in the appendix.
* This approach has a nice property that the runtime looks quite efficient (unless the alignment is really slow). With space, it would be nice to compare the asymptotic runtime of the full algorithm vs. some of the more recent models like Bohnet et al., 2023, Liu et al., 2022, and Wu et al., 2020, and Paolini et al., 2021.
* L617-624: I believe in OntoNotes, generic mentions are not intentionally treated as *not coreferring*. It’s worth double-checking in their guidelines, but then this example about “students” should not be used as an annotation error.
* There's something about this writing style/tone that reads a little arrogant/overconfident, especially given some of the questions/comments in my review -- specifically the final line "Our model’s strong results fundamentally challenge the default task-specific approach to coreference resolution." even though prior work has aimed at non-task-specific models (e.g. Toshniwal et al., 2022's work that trained on multiple datasets, or Bohnet et al., 2023's work on a single multilingual model). Other examples of overconfident statements in intro/abstract, e.g. by claiming size is the main factor or that the model actually is not "task-specific."

---

> ### Author Rebuttal · Authors · 2023-08-28
>
> We would like to thank the reviewer for the positive review and constructive comments.
>
> **Responses to the questions**
> - A.  T5-3B is substantially better than T5-large (77.2), and they are trained on the same C4 dataset. Model size becomes particularly significant when it is not adequately large. However, once the model size reaches or exceeds 3 billion parameters, further enlarging it yields diminishing returns as performance plateaus.
> - B.
>   - The main hypothesis for why seq2seq now works is that the model size is now sufficiently large. T5-3B achieves substantial improvements over T5-large, which is pretrained on the same C4 dataset.
>   - Thanks for your valuable suggestion! We will include a separate discussion section for the hypotheses in the final version.
> - C.
>   - Two special tokens, "<sentence>" and "</sentence>", serve as sentence boundary markers, indicating the start and end of each sentence respectively in both the source and target sequences. During generation at inference time, the model determines the sentence index where the next token will belong by keeping track of the number of “<sentence>” and “</sentence>” tokens generated so far. The constrained decoding restricts the model to generating tokens that correspond to the same sentence index in the source sequence, as well as coreference-specific special tokens like "<m>", "</m>", "|", and integers. We will include the details of constrained decoding rules for all the linearization and action sequence variants in the Appendix in the final version.
>   - If oracle alignment were available, the final score would likely improve. However, it is doubtful that this would surpass the performance of the full linearization variants, given the reduced contextual information in the target sequence. The examination of attention weights in the final layer is really an interesting approach for alignment. Thanks for proposing that!  We will include this in the final version.
> - D.  The task-specific in our paper means that our approach is not task-specific since seq2seq is a general approach that can be used for many tasks. The specifically trained model is definitely task-specific. We are not aiming to train a general model like ChatGPT that can solve many tasks. Within the coref task, we trained dataset-specific coref models as well as a joint multi-dataset coreference model with our method. Table 3 presents the performance. The OntoNotes performance  (82.9) for the joint model will be included in the final version. When assessed on each respective training dataset, the multi-dataset joint model either surpasses or matches the performance of dataset-specific models.This shows the possibility of training a general coref model on sufficient training datasets with our approach.
> - E. CorefQA uses extra training data specifically for mention proposal training, a crucial element of their approach. This stage is more appropriately considered as additional fine-tuning rather than pretraining, which is commonly associated with general language model training. This distinction is a primary reason why recent studies employing large-scale models like 3B/3.7B/11B/13B (m)T5/T0, such as ASP, Bohnet et al 2023, and our work, often regard CorefQA as not directly comparable. Furthermore, CorefQA remains unreproducible. We plan to make all our code available to ensure reproducibility.
> - F. Due to the autoregressive generation process, if k clusters has been identified prior to a given step, then k+1 special mention end boundary tokens ("</m1>", "</m2>", ..., "</mk>", "<new>") become selectable options. If a document contains 100 clusters and all have been identified prior to a generation step, 100 special "</m_l>" mention end boundary tokens will be available for selection. No recency bias is applied when choosing among these 100 tokens.
>
> **Presentation improvements**
>
> - We will make a footnote that comparison is still not entirely equivalent for Bohnet et al., 2023.
> - We will release all the code for reproducibility. We will also include the constrained decoding details in the Appendix in the final version.
> -  Ideally, we would also obtain the inference speed of baseline models by running them under identical settings to ours. However, this is not feasible as those baseline models either lack fully reproducible code, as is the case with Bohnet et al., 2023 and Wu et al., 2020, or their implementation is inefficient, as seen in Liu et al., 2022, which employs a rudimentary model parallel technique instead of a more efficient approach like Deepspeed we use.
> - Thanks for pointing out the OntoNotes annotation guidelines! We will fix this mistake by using this replacement example: The gold annotation omitting “he” and “she” is “as [[ William ]9 and  she ]10  saw each other, it was such a wonderful reunion for [ them ]10 to just hug, and he would hug [ her ]2 and look at [ her ]2 ” while our model correctly predicts  “as [ [ William ]11 and [ she ]2 ]12 saw each other, it was such a wonderful reunion for [ them ]12 to just hug, and [ he ]11 would hug [ her ]2 and look at [ her ]2”.
> - We will tone down the claims in the future version.

---

### Official Review · Reviewer_SffD · 2023-08-05

**Soundness:** 4

**Excitement:**

4: Strong: This paper deepens the understanding of some phenomenon or lowers the barriers to an existing research direction.

**Paper Topic And Main Contributions:**

This paper presents a supervised method for directly applying pre-trained sequence-to-sequence models to the problem of entity coreference resolution without the need for task-specific architectural components. The method functions by taking the input sequence to be the raw string of tokens composing a document, and the output sequence to be the document with special tokens marking entity boundaries and cluster identities. For example, the input sequence might be the tokenization of the string “Emily thinks she is tall” and the corresponding output sequence would then be the tokenization of “<m> Emily | 1 </m> thinks <m> she | 1 </m> is tall” where the integers after each mention indicate the cluster identity. The output sequence is decoded one token at a time by concatenating the previous output to the input sequence.

The paper explores possible decisions for how to represent mention’s cluster identity and how to represent the output sequence. For the output sequence, the paper shows that replacing non-special tokens with a “copy” token performs better than other methods. The paper experiments training the proposed model on OntoNotes, PreCo, LitBank, and a joint dataset of all three. For OntoNotes, the paper shows that finetuning T0-3B (3 billion parameters) and T0-pp (11 billion parameters) using the proposed method outperform existing methods. The best configuration achieves a CoNLL F1 score of 83.2 on OntoNotes.

**Questions For The Authors:**

A) L219: What specifically do you mean by a single pass?

B) L327: What is the reason for choosing 0.0001?

C) Table 1: are the values for # clusters correct? These are described as the average number of clusters in a document but e.g. PreCo has significantly more than 1.6 entity clusters per document. Maybe this is the average mentions per cluster?

D) Table 1: from the numbers it appears that you are not using the original PreCo test set. How exactly did you split the dev and test splits for PreCo?

E) What is the reason for using a batch size of 1 per GPU? Is this to accommodate the largest possible maximum input length?

F) How long does it take to train the model in terms of iterations and wall-clock time? For the inference times given in minutes is this using the same number of GPUs as training?

**Reasons To Accept:**

* The proposed method achieves state-of-the-art results in all four training configurations including when trained on OntoNotes, the most well studied entity coreference evaluation.
* The paper includes extensive baselines including training the seq2seq model of Paolini et al. (2021) with a larger T0-3B language model.
* The writing is very clear and the methodology is convincing.
* The paper shows the impact of various design decisions at a large scale (finetuning T0-3B) through both ablations and the comparison of proposed modeling decisions (partial linear vs full linear, token action vs copy action, integer vs integer free).

**Reasons To Reject:**

* The paper is framed as providing evidence that task-specific models are not necessary for achieving state-of-the-art performance on coreference resolution; however, this finding is not surprising given the prior work of Bohnet et al. (2023) which achieved state-of-the-art performance on entity-based coreference resolution datasets, including OntoNotes, by formulating coreference resolution as a sequence-to-sequence task and finetuning T5 and MT5 on said formulation. Bohnet et al. (2023) was published in TACL on 14 March 2023 which was more than 3 months prior to the EMNLP submission deadline of 23 June 2023. My understanding is that this would therefore not be considered contemporaneous work as per the EMNLP 2023 CfP. The paper does compare against Bohnet et al. (2023), but this comparison should be more extensive to support the claim that the proposed method is not task-specific whereas Bohnet et al. (2023) is considered in the paper to be task-specific.
* The provided error analysis is very brief with limited insight into model performance beyond the observation that mention detection is a bottleneck.
* The evaluation is largely focused on F1 score as opposed to other dimensions of performance such as inference speed and model size. Based on Table 6, it appears that at comparable model sizes, task-specific models significantly outperform the proposed method.

**Reproducibility:**

3: Could reproduce the results with some difficulty. The settings of parameters are underspecified or subjectively determined; the training/evaluation data are not widely available.

**Reviewer Confidence:**

4: Quite sure. I tried to check the important points carefully. It's unlikely, though conceivable, that I missed something that should affect my ratings.

**Typos Grammar Style And Presentation Improvements:**

* In the Error Analysis, I do not believe the “annotation mistakes” are necessarily mistakes but rather unintuitive results of the complexity of coreference annotation. For example, bare plurals in OntoNotes such as “students” are not annotated as coreferring except in certain cases such as when the antecedent of a pronoun which then starts a new coreference chain. While unintuitive, this example does appear to follow the OntoNotes annotation guidelines.
* L464: The wording of “large T5” is confusing since there is a large-size version of the T5 model.
* Table 2: It took me a while to figure out of the 5 models which action or linearization were being used when not specified.

---

> ### Author Rebuttal · Authors · 2023-08-28
>
> We thank the reviewer for the review and comments.
>
> **Responses to concerns**
>
> - In response to the reviewer's questions regarding the task-specific claims and our comparison with Bohnet et al, 2023, we clarify why we categorize Bohnet et al, 2023 as a task-specific model and outline the core distinctions between our methodology and that of Bohnet et al, 2023, as follows:
>     - The “model” in our claim refers to both the model architecture and the system design.  The method of Bohnet et al, 2023 is considered task-specific because of its task-specific system design, while all other baseline models are considered task-specific because of their unique task-specific model architecture. Our method doesn’t require any coref-specific architecture or system. All we need is a seq2seq model, which is general and task-agnostic, having been widely employed in a diverse array of tasks like machine translation and text summarization, which can be very different from each other by nature. Bohnet et al, while not needing a task-specific architecture, does necessitate a task-specific transition-based system, which is specifically designed and adapted for the coreference task described in the paper.  Due to the nature of coreference resolution that clusters mentions, the “Link” and “Append” actions in the transition-based seq2sq system are suitable and effective to cluster mentions together, that’s why it’s effective and achieves strong performance on coreference resolution. The success of the transition system of Bohnet et al. does not imply the success of the general seq2seq system in our work, because the transition system is explicitly designed to solve the mention clustering problem, whereas the general seq2seq system is not.
>    - The core of the transition-based seq2seq model is the transition-based system, not the seq2seq model itself. In their paper, the seq2seq model serves to verbalize their coref-specific transition-based system. In contrast, our approach uses a standard seq2seq model to address coreference resolution by predicting a coreference annotation of the text directly.
>
> - We will strengthen the error analysis section by looking into more error patterns in the final version.
> -  It’s well known that Seq2seq models, such as T5, necessitate an adequate model size for satisfactory performance. This observation is consistent across recent coreference papers, including Liu et al., 2022 and Bohnet et al, 2023.  However, once the model size reaches sufficient capacity, further enlarging it yields diminishing returns as performance plateaus. Despite scaling up task-specific models with ample computational resources, it's unlikely that they would surpass the performance of our approach or recent models based on sizeable (m)T5.
> - Our paper incorporates data on our model's inference speed. Ideally, we would also obtain the inference speed of baseline models by running them under identical settings to ours. However, this is not feasible as those baseline models either lack fully reproducible code, as is the case with Bohnet et al., 2023 and Wu et al., 2020, or their implementation is inefficient, as seen in Liu et al., 2022, which employs a rudimentary model parallel technique instead of a more efficient framework like Deepspeed we use.
>
> **Responses to questions**
> - A)  Our model runs in a single pass for each document, generating a target sequence all at once based on the entire input document. In contrast, Bohnet et al's approach processes one sentence at a time from a raw document, manually augmenting it for next sentences. For each raw sentence, the model predicts "Link" and "Append" actions based on that sentence, then manually augments it with the prediction. The augmented sentence is then used as history context for the following conditional generation.
> - B) The 0.0001 is a hyperparameter. We didn’t explore this much since we found the performance is stable and good when it’s small (<=0.0001). We’ll indicate it as a hyperparameter in the writing for the final version.
> - C) Thanks for correcting this mistake! It’s actually the average mentions per cluster. We will fix this in the final version.
> - D) We follow the setting of Toshniwal et al (https://aclanthology.org/2021.crac-1.12.pdf) and use the last 500 docs of the original training data as dev and the original dev data as test.
> - E) Yes, this is to accommodate the largest possible maximum input length.
> - F) The training time of 3 billion models is around 20 hours on 8 A100 40G GPUs and the training time of the 11 billion models is around 36 hours on 8 A100 80G GPUs.  The inference uses the same number of GPUs as training.
>
> **Presentation improvements**
> - Thanks for pointing out the OntoNotes annotation guidelines! We will fix this error in the future version by using this replacement example: The gold annotation omitting “he” and “she” is “as [[ William ]9 and  she ]10  saw each other, it was such a wonderful reunion for [ them ]10 to just hug, and he would hug [ her ]2 and look at [ her ]2 ” while our model correctly predicts  “as [ [ William ]11 and [ she ]2 ]12 saw each other, it was such a wonderful reunion for [ them ]12 to just hug, and [ he ]11 would hug [ her ]2 and look at [ her ]2”.
> - We will change the “large T5” wording to “sizeable T5” in the final version.
> - We will make Table 2 more clear in the final version.

---

### Official Review · Reviewer_Jhxb · 2023-08-12

**Soundness:** 3

**Excitement:**

3: Ambivalent: It has merits (e.g., it reports state-of-the-art results, the idea is nice), but there are key weaknesses (e.g., it describes incremental work), and it can significantly benefit from another round of revision. However, I won't object to accepting it if my co-reviewers champion it.

**Paper Topic And Main Contributions:**

This paper presents a seq2seq transition-based reduction of entity coreference resolution (ECR) that the authors claim performs better than SOTA or at least matches it on numerous datasets. Furthermore, the authors conduct empirical analyses to theorize the impact of different model attributes and sequence generation strategies on coreference performance, employing a series of ablation experiments.

**Reasons To Accept:**

- A working "linear" seq2seq approach for ECR that beats SOTA and yet is simple
- experiments on a broad range of datasets
- a solid ablation study

**Reasons To Reject:**

- The intro is very confusing without diagrams or examples. In general, the paper would read a lot better with fewer numbers
- The seq2seq approach is fundamentally similar to Bohnet et al., 2023. The authors only provide contributions by simplistic variations of the decoding strategies and model attribute variations.
- While the results on a spectrum of models of different sizes were interesting, an error analysis focusing on the gaps between the smallest and largest models would have been really useful. The error analysis in this paper seems limited.
- The authors do not provide supplemental code to back the claims of the paper, given that the main contribution of this paper is an engineering experiment (could consider using anonymous GitHub)

**Reproducibility:**

3: Could reproduce the results with some difficulty. The settings of parameters are underspecified or subjectively determined; the training/evaluation data are not widely available.

**Reviewer Confidence:**

4: Quite sure. I tried to check the important points carefully. It's unlikely, though conceivable, that I missed something that should affect my ratings.

**Typos Grammar Style And Presentation Improvements:**

- lines 062-074: extremely difficult to follow without an example. I would suggest adding a full system architecture diagram with an example.
- lines 075-096: too many numbers makes it really difficult to read. Maybe you can re-write it without the result numbers and summarize the key contributions (or the takeaways in your case)
- lines 288, 289, 298, 299 might read better without the commas like in 217
- Table 2 has too many numbers. Since you only talk about the CoNLL F1 in Results, it might be better if you move this table to appendix and use a condensed table with CoNLL F1 only. The freed up space could be then utilized for a better presentation of the system.
- It would also be worth adding the LEA score if you want to talk about the precision and recall too specifically in your ablation study and error analysis.
- Table 6 might look better and more informative as a plot (# params on x, performance on y)

---

> ### Author Rebuttal · Authors · 2023-08-28
>
> We would like to thank the reviewer for the review and suggestions.
> -  Our method is fundamentally different from Bohnet et al., 2023. The core of the transition-based seq2seq model is the transition-based system, not the seq2seq model itself. In their paper, the seq2seq model serves to verbalize their coref-specific transition-based system. In contrast, our approach uses a standard seq2seq model to address coreference resolution by predicting a coreference annotation of the text directly.  The success of the transition system of Bohnet et al. does not imply the success of the general seq2seq system in our work, because the transition system is explicitly designed to solve the mention clustering problem, whereas the general seq2seq system is not.
> - We will release all our code.
> - We will incorportate the writing suggestions in the final version.

---

### Meta-Review · Area_Chair_iGkS · 2023-09-08

**Recommendation:** 4

**Metareview:**

The paper presents a seq2seq approach to coreference resolution. Unlike Bohnet et al (2023) which use a transition-based approach to processing a sentence at a time. The proposed approach directly predicts the documents with intext coreference annotations. The evaluation of a number of corpora showed results matching or exceeding the previous state-of-the-art results. The paper also reports results on the smaller LMs which is useful for researchers who do not have access to the resources needed for the large models. The main issue of the paper, however, is they made some strong claims to oversell the paper, e.g. claims on Bohnet et al (2023) are task-specific system etc. Apart from this, I would also expect the author to make the code freely available, and address other issues raised by the reviewers. A discussion on post-processing is also needed, given the LMs might not always generate texts following the input text. How you align the input and output can be a piece of useful information, e.g. Bohnet et al used the 3-word immediately after the mention to find the position of the mention.

---

### Decision · Program_Chairs · 2023-10-07

**Decision:**

Accept-Main

**Comment:**

The paper presents a seq2seq approach to coreference resolution. Unlike Bohnet et al (2023) which use a transition-based approach to processing a sentence at a time. The proposed approach directly predicts the documents with intext coreference annotations. The evaluation of a number of corpora showed results matching or exceeding the previous state-of-the-art results. The paper also reports results on the smaller LMs which is useful for researchers who do not have access to the resources needed for the large models. The main issue of the paper, however, is they made some strong claims to oversell the paper, e.g. claims on Bohnet et al (2023) are task-specific system etc. Apart from this, I would also expect the author to make the code freely available, and address other issues raised by the reviewers. A discussion on post-processing is also needed, given the LMs might not always generate texts following the input text. How you align the input and output can be a piece of useful information, e.g. Bohnet et al used the 3-word immediately after the mention to find the position of the mention.